# Understanding activity trends in electrochemical water oxidation to form hydrogen peroxide

Xinjian Shi [1,2], Samira Siahrostami[1], Guo-Ling Li [3,4], Yirui Zhang[2,5], Pongkarn Chakthranont[1], Felix Studt[3,6,7], Thomas F. Jaramillo [1], Xiaolin Zheng [1,2] & Jens K. Nørskov[1,3]

Electrochemical production of hydrogen peroxide ($H_2O_2$) from water oxidation could provide a very attractive route to locally produce a chemically valuable product from an abundant resource. Herein using density functional theory calculations, we predict trends in activity for water oxidation towards $H_2O_2$ evolution on four different metal oxides, i.e., $WO_3$, $SnO_2$, $TiO_2$ and $BiVO_4$. The density functional theory predicted trend for $H_2O_2$ evolution is further confirmed by our experimental measurements. Moreover, we identify that $BiVO_4$ has the best $H_2O_2$ generation amount of those oxides and can achieve a Faraday efficiency of about 98% for $H_2O_2$ production.

[1] SUNCAT Center for Interface Science and Catalysis, Department of Chemical Engineering, Stanford University, 443 Via Ortega, Stanford, CA 94305, USA. [2] Department of Mechanical Engineering, Stanford University, Stanford, CA 94305, USA. [3] SUNCAT Center for Interface Science and Catalysis, SLAC National Accelerator Laboratory, Menlo Park, CA 94025, USA. [4] School of Physics and Engineering, Henan University of Science and Technology, Luoyang 471023, China. [5] Department of Mechanical Engineering, Tsinghua University, Beijing 100084, China. [6] Institute of Catalysis Research and Technology, Karlsruhe Institute of Technology, 76344 Eggenstein-Leopoldshafen, Germany. [7] Institute for Chemical Technology and Polymer Chemistry, Karlsruhe Institute of Technology, 76131 Karlsruhe, Germany. Xinjian Shi, Samira Siahrostami and Guo-Ling Li contributed equally to this work. Correspondence and requests for materials should be addressed to X.Z. (email: xlzheng@stanford.edu) or to J.K.Nør. (email: norskov@stanford.edu)

Hydrogen peroxide ($H_2O_2$) is an important chemical with a wide range of applications in industry, such as paper and textile manufacturing and environmental protection for detoxification and color removal of wastewater[1]. Currently, $H_2O_2$ is produced indirectly via the anthraquinone oxidation process, which is an energy-demanding multi-electron process and requires large plants[1]. Moreover, transportation of $H_2O_2$ to the place of use adds additional challenges due to safety concerns. Broader usage of $H_2O_2$ could benefit from the capability of direct on-site production. Electrochemical synthesis of $H_2O_2$ provides a straightforward route for on-site production and ideally solves the issues associated with the indirect anthraquinone route[2–9]. One of the attractive possible routes for electrochemical $H_2O_2$ generation is via two-electron oxidation of water (Eq. 1)[10–12]:

$$2H_2O \rightarrow H_2O_2 + 2(H^+ + e^-) \quad E^\circ = 1.76\,V \quad (1)$$

This process (Eq. 2) is desirable since it can be coupled with hydrogen evolution reaction (Eq. 2) to simultaneously produce two valuable products: $H_2O_2$ and $H_2$ (Eq. 3) in a single electrochemical device using only water as raw material. Such a device can also be coupled with photoabsorbers to utilize sunlight for both reactions:[12–14]

$$2(H^+ + e^-) \rightarrow H_2 \quad E^\circ = 0.0\,V \quad (2)$$

$$2H_2O \rightarrow H_2O_2 + H_2 \quad (3)$$

However, the two-electron water oxidation (Eq. 1) must compete with the four-electron oxidation reaction for $O_2$ generation (Eq. 4) and the one-electron oxidation reaction for producing OH radical (Eq. 5)[12].

$$2H_2O \rightarrow O_2 + 4(H^+ + e^-) \quad E^\circ = 1.23\,V \quad (4)$$

$$H_2O \rightarrow OH^\bullet(aq) + (H^+ + e^-) \quad E^\circ = 2.38\,V \quad (5)$$

The relevant intermediates of the one- (Eq. 5[12]), two- (Eq. 1) and four-electron (Eq. 4) water oxidation reactions are OH*, O* and OOH*. Historically, the $O_2$ generation reaction (Eq. 4) has been the main focus of the water oxidation research[15–29] and less attention has been paid to the selective two-electron oxidation reaction of water to $H_2O_2$ (Eq. 1), which is considered as a much more difficult process. A main challenge in realizing such a photoelectrochemical production of $H_2O_2$ system is to find a material that can selectively and efficiently produce $H_2O_2$ from water. Electrochemical oxidation of water has been reported over various metal oxides, such as $MnO_x$[10, 11], $WO_3$-$BiVO_4$[14, 30, 31] and $TiO_2$[32–36]. These pioneering works have not only demonstrated the potential of $H_2O_2$ production over metal oxides but also suggested that $BiVO_4$ is the best oxide for $H_2O_2$ production[14, 31]. Nevertheless, little is known about the energy barrier and the limiting potential for $H_2O_2$ generation for different metal oxides. Even less is known about how the $H_2O_2$ generation efficiency of these metal oxides vary with applied biases both theoretically and experimentally. Such study on the bias-dependent $H_2O_2$ production is critical for understanding the competition among one-, two- and four-electron pathways under different bias, a prerequisite for identifying the optical bias range for $H_2O_2$ generation.

In the present work, we theoretically investigate the activity trends of four different oxides (i.e., $WO_3$, $SnO_2$, $TiO_2$ and $BiVO_4$) towards water oxidation for $H_2O_2$ production with further experimental validation. Both the calculated and measured onset potentials for $H_2O_2$ production increase in the sequence of $WO_3$, $BiVO_4$, $SnO_2$ and $TiO_2$. Among all these four oxides, $BiVO_4$ is

identified as the best catalyst for the two-electron water oxidation in dark and under illumination, and this result is consistent with previous studies on comparing different metal oxides for $H_2O_2$ production[14, 31]. Importantly, we identify the optimal bias range for $BiVO_4$ to produce $H_2O_2$ in dark (~2.9–3.3 V vs RHE) and under illumination (~1.7 V–2.3 V vs RHE). As such, $BiVO_4$ achieves a high faraday efficiency (FE) of 70% in dark and 98% under 1 sun illumination.

## Results

**Theoretical analyses.** As discussed above, $H_2O_2$ synthesis from water oxidation is a challenging reaction. This is due to the fact that selectivity and activity of the materials are largely limited by several criteria imposed by the thermodynamics of the competing reactions[13]. The adsorption free energies of relevant intermediates of the one- (Eq. 5), two- (Eq. 1) and four-electron (Eq. 4) water oxidation reactions, i.e., OH*, O* and OOH* can be calculated using density functional theory (DFT). We show here that the free energies of OH* and O* are key parameters determining the selectivity and activity towards different oxidation products, $O_2$ (Eq. 4), OH radical (Eq. 5) or $H_2O_2$ (Eq. 1)[13]. Using DFT, we calculated the free energies of OH*, O* and OOH* on $BiVO_4$ (details of calculations in Supplementary Note 1). We only focus on the (111) surface, which has been shown theoretically and experimentally to be stable and exposed in the $BiVO_4$ crystal structure[37]. In addition, we have taken the OH*, O* and OOH* free energies for $WO_3$(100), $TiO_2$(110) and $SnO_2$(110) from reported DFT calculations (Supplementary Note 1, Supplementary Fig. 1 and Supplementary Tables 1, 2)[13, 38–40]. We use the computational hydrogen electrode (CHE) model, which exploits that the chemical potential of a proton–electron pair is equal to gas-phase $H_2$ at standard conditions. The electrode potential is taken into account by shifting the electron energy by $-eU$, where $e$ and $U$ are the elementary charge and the electrode potential, respectively[41]. The limiting potential for the electrochemical reaction to occur is defined as the lowest potential, at which all the reaction steps are downhill in free energy following the previous report[13]. Figure 1 shows the activity volcano plots based on the calculated limiting potentials as a function of the calculated free energy of OH* ($\Delta G_{OH^*}$) for both two-electron (black) and four-electron (blue) oxidation reactions.

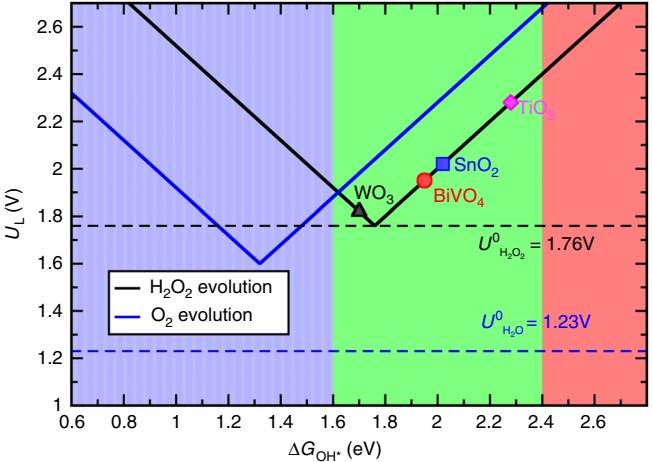

**Fig. 1** Activity volcano plots. It is based on calculated limiting potentials as a function of calculated adsorption energies of OH* ($\Delta G_{OH^*}$) for the two-electron oxidation of water to hydrogen peroxide evolution (*black*) and the four-electron oxidation to oxygen evolution (*blue*). The corresponding equilibrium potentials for each reaction have been shown in *dashed lines*

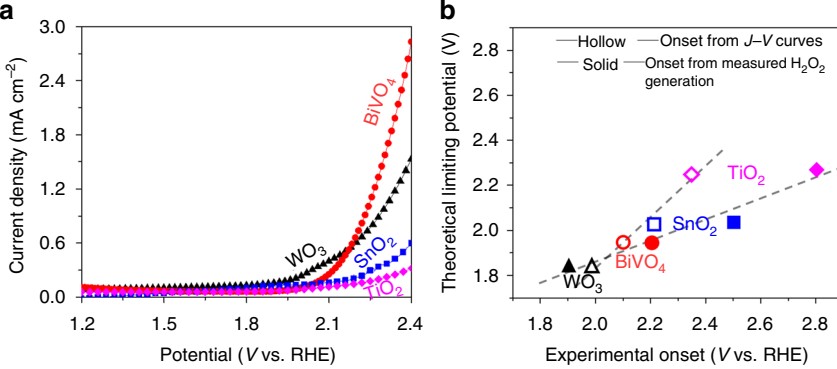

**Fig. 2** $J$–$V$ curves and onset potentials. **a** $J$–$V$ curves of four metal oxides without illumination, for which the current onset suggest the onset of $H_2O_2$ production. **b** Theoretical predicted onset potentials vs experimental measured onset potentials for $H_2O_2$ production. The values on the y-axis are the theoretical limiting potential obtained from Fig. 1. As to the x-axis, the values of hollow points were the potentials for the $J$–$V$ curves of different metal oxides to reach 0.2 mA cm$^{-2}$ in Fig. 2a, while the values of the solid points were the potentials at which the generated $H_2O_2$ concentration reaches 1 ppm, by measuring a 1 cm$^2$ sample in the 20 ml electrolyte for 10 min

The corresponding equilibrium potentials for each reaction have been shown in dashed lines.

From the thermodynamic point of view, materials with strong OH adsorption energy (shaded in blue in Fig. 1) will further oxidize OH* to O* and OOH*, following the complete four-electron oxidation reaction (Eq. 4) to evolve oxygen. Therefore, electrocatalysts with weak OH* free energy will have low selectivity towards the four-electron pathway but high preference towards the two-electron route. At the same time, the OH* free energy should be strong enough to dissociate the water molecule and provide a good thermodynamic driving force, for the two-electron pathway towards $H_2O_2$. The free energy for $H_2O_2$ formation is ~3.5 eV, twice of the equilibrium potential for Eq. 1, so the electrocatalyst should have $\Delta G_{O*} \gtrsim 3.5$ eV. Given the fact that the *O and *OH energies are generally found to scale ($\Delta G_{O*} = 2\Delta G_{OH*} + 0.28$)[38], this sets a lower limit for OH* free energy, i.e., $\Delta G_{OH*} \gtrsim \frac{3.5}{2} - \frac{0.28}{2} \sim 1.6$ eV. The upper limit for $\Delta G_{OH}$ is set by the free energy of OH radical formation in the solution (Eq. 5), since too weak OH* free energy with $\Delta G_{OH*} >$ ~2.4 eV drives the reaction towards OH radical formation (pink shaded area in Fig. 1). Hence, the combined thermodynamic criteria and scaling relation indicates a selective catalyst for $H_2O_2$ evolution should have $\Delta G_{OH*}$ from ~1.6 to 2.4 eV. This thermodynamic analysis suggests that $WO_3$, $SnO_2$, $BiVO_4$ and $TiO_2$ should be able to generate $H_2O_2$ within certain values of the OH* free energy (shaded in green in Fig. 1). To increase the selectivity region for $H_2O_2$ evolution, we need to identify catalyst materials that largely deviate from the O* and OH* scaling relation[42].

Aside from the high selectivity, the two-electron oxidation reaction (Eq. 1) ideally should also have high activity with low overpotential. The theoretical overpotential is defined as the difference between the limiting potential and equilibrium potential (1.76 V for the two-electron path). The overpotential is governed by the binding of OH* to the catalyst surface, so controlling the overpotential is a matter of tuning the free energy of OH*[41]. An OH* free energy ($\Delta G_{OH*}$) of 1.76 eV, when calculated at zero potential and relative to liquid water, will give zero overpotential. The calculated theoretical limiting potential for $BiVO_4$ is 1.95 V, hence it has a theoretical overpotential of ~0.2 V for the two-electron oxidation reaction. The activity of $BiVO_4$ can be further improved with different doped metals such as Sr and Ru (Supplementary Note 2, Supplementary Fig. 2 and Supplementary Table 3). The calculated limiting potentials for $WO_3$, $SnO_2$ and $TiO_2$ are 1.82, 2.02 and 2.27 V, respectively. In the following, we show that the trend in theoretical limiting

potentials for $WO_3$, $BiVO_4$, $SnO_2$ and $TiO_2$ is in very good agreement with experimental measurements.

**Materials fabrication**. Experimentally, we evaluate the $H_2O_2$ evolution performance of four oxides: $WO_3$, $BiVO_4$, $SnO_2$ and $TiO_2$ by determining their onset potentials, faraday efficiencies and production rates of $H_2O_2$ per geometric area of electrodes. All the oxides were synthesized on transparent and conductive fluorine-doped tin oxide (FTO) substrates. The $WO_3$ was synthesized by flame vapor deposition (FVD). $BiVO_4$, $SnO_2$, and $TiO_2$ were synthesized by a sol–gel process (see Methods and Supplementary Note 3). Since the electrochemical performance of each oxide is affected by its loading, each oxide film was individually optimized to yield the highest FE for the dark electrochemical measurement before taken for comparison. For example, Supplementary Fig. 3 shows that the measured FE for $H_2O_2$ production varies with the $BiVO_4$ loading, which was controlled by varying the precursor concentration during the spin coating process (Supplementary Fig. 4). The low loading results in low $BiVO_4$ coverage and exposed FTO, leading to a low $H_2O_2$ production. On the other hand, the high coverage results in high film resistance that impedes the charge transport process. Hence, for each oxide investigated, its loading on FTO was individually optimized for $H_2O_2$ production and the dependence of the FE on sample preparation conditions for each oxide is listed in Supplementary Table 4.

**$H_2O_2$ production comparison under dark conditions**. Next, we measured the current–voltage ($J$–$V$) curves for the four oxides ($WO_3$, $BiVO_4$, $SnO_2$ and $TiO_2$) without illumination (Fig. 2a). The $J$–$V$ onset, as well as the experimental measured $H_2O_2$ evolution onset potentials (defined as the potential at which the $H_2O_2$ concentration reaches 1 ppm), are compared with the calculated theoretical potentials in Fig. 2b. It can be seen that both the measured current onset potential (hollow symbols) and the onset potential for $H_2O_2$ generation (solid symbols) increase in the order of $WO_3 < BiVO_4 < SnO_2 < TiO_2$, which agrees with the theoretical predication, supporting the validity of using $\Delta G_{OH*}$ as a descriptor to analyze the $H_2O_2$ evolution onset (Fig. 1). The measured onset potentials are higher than the calculated values, which is likely due to the additional kinetic barriers to be overcome in the actual experiments.

We further quantified the FE and the amount of $H_2O_2$ generated from the oxides as functions of the applied biases. As shown in Fig. 3a, all the metal oxides investigated share the

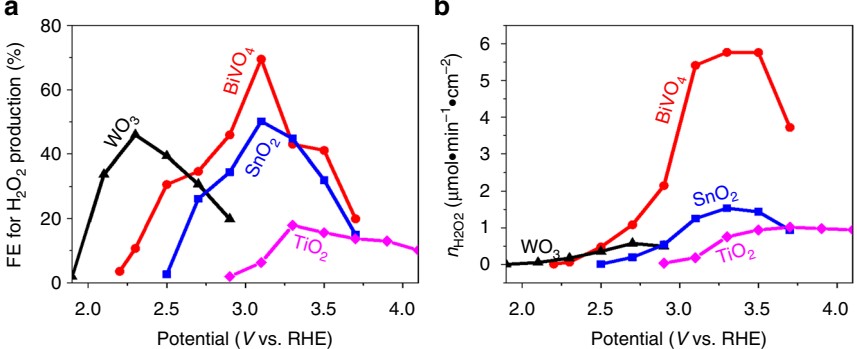

**Fig. 3** The faraday efficiency (FE) and mole amount of $H_2O_2$ under dark. **a** The FE and **b** the mole amount ($n$) of $H_2O_2$ generation ($n$) vs potential ($V$) without illumination. Both show that $BiVO_4$ has the highest FE and $n$ for $H_2O_2$ production over other metal oxides

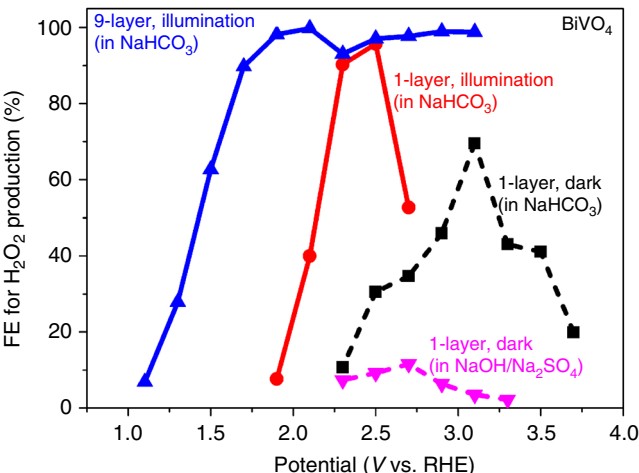

**Fig. 4** The FE for $BiVO_4$ under both dark and illumination. FE vs applied bias for $BiVO_4$ under different conditions, including different electrolytes in darkness, same electrolyte between darkness and illumination, and different layers under illumination. The thicker $BiVO_4$ with light illumination in $NaHCO_3$ shows best performance for $H_2O_2$ production

similar FE trend with increasing applied bias: the FE first increases to a maximum value and then decreases. Or equivalently, for each metal oxide, there is an optimal potential window for $H_2O_2$ production, due to the competition with one-electron and four-electron oxidation reactions as predicted by theory (Fig. 1). Among all the oxides investigated, $BiVO_4$ achieves the highest FE of 70% at 3.1 V vs RHE. In addition, $BiVO_4$ also has the highest $H_2O_2$ production rate per geometric area of electrode (Fig. 3b), due to the combined high FE (Fig. 3a) and high current density (Fig. 2a).

**$H_2O_2$ production on $BiVO_4$ under different conditions**. The above results clearly indicate that $BiVO_4$, among the four oxides investigated, has the best electrochemical properties towards $H_2O_2$ production, which is consistent with the previous screening studied of different metal oxides for $H_2O_2$ generation[31]. Next, we further investigated the influence of other conditions on the $H_2O_2$ production of using $BiVO_4$, including the electrolyte, light illumination and sample thickness. The two dash lines in Figure 4 are the measured FEs for the thin 1-layer $BiVO_4$ (i.e., spin coating once) under dark using two different electrolytes: 1 M $NaHCO_3$ (pH = 8.3) and 0.5 M $Na_2SO_4$ (adjusted to pH = 8.3 with NaOH). Clearly, $NaHCO_3$ is a much better electrolyte than NaOH/$Na_2SO_4$ for the $H_2O_2$ production from water oxidation. This

observation is consistent with recent work by Fuku et al.,[14] which reported that $HCO_3^-$ is beneficial for the $H_2O_2$ production. In addition, as shown in Fig. 4, illumination not only shifts the onset potential to lower values but also increases the FEs for $H_2O_2$ production from 70% to over 95% for the 1-layer $BiVO_4$ in $NaHCO_3$. The reasons are that illumination introduces additional photogenerated charge carriers for $H_2O_2$ production and supplies photovoltage to allow $H_2O_2$ generation from water oxidation at a lower external bias[43]. To further enhance the benefits under illumination, we increase the loading of $BiVO_4$ by spin coating nine times (referred as 9-layer $BiVO_4$) to enhance the light absorption. The increased thickness for $BiVO_4$ further shifts the onset potential for $H_2O_2$ to less than 1.1 V vs RHE, which is over 1.1 V lower than that of the dark conditions. This lowered onset potential and increased FEs strongly support that $BiVO_4$ is a promising photoanode material for the $H_2O_2$ production in a photoelectrochemical system. The $J$–$V$ curve and the measured $H_2O_2$ generation rate under illumination for this 9-layer $BiVO_4$ are shown in Supplementary Fig. 5. In addition, we have measured the evolved gaseous $O_2$ and $H_2$ and the measured FEs are shows in Supplementary Fig. 6. The figure shows that the sum of the FE ($O_2$) and FE ($H_2O_2$) is about 98–103%, confirming the accuracy in our $H_2O_2$ concentration measurement. Finally, the photoelectrochemical stability of $BiVO_4$ is known to be an issue when the electrolyte is far from neutral conditions because the $V^{5+}$ tends to dissolve into solution[44, 45]. However, we used the bicarbonate electrolyte with a measured pH value of 8.3; hence, $BiVO_4$ is relatively stable in this near neutral region[46].

**Discussion**

In the present work, we utilized DFT calculations in conjunction with experimental measurements to study the activity of two-electron water oxidation towards $H_2O_2$ evolution over four metal oxides, namely $WO_3$, $BiVO_4$, $SnO_2$ and $TiO_2$. Both the calculated and measured onset potentials for $H_2O_2$ production increase in the sequence of $WO_3$, $BiVO_4$, $SnO_2$ and $TiO_2$. Among all these four oxides, $BiVO_4$ produces the highest faraday efficiency (~70%) and largest amount for $H_2O_2$ under dark. The peak faraday efficiency $BiVO_4$ is further increased to 98% by adding illumination, optimizing electrolyte and optimizing the thickness of $BiVO_4$. Those optimizations also lower the onset potential from 2.2 V to <1.1 V. These results suggest that $BiVO_4$ is an excellent photoanode candidate for electrochemical and photoelectrochemical $H_2O_2$ production. The theoretical simulation and experimental demonstration illustrated in this work have furthered the understanding of the activity and selectivity of water oxidation to $H_2O_2$ on metal oxide surfaces. Our result has opened an avenue for novel photoelectrochemical device designs with

fundamental mechanism study that utilize solar energy and water to produce an oxidative product with higher value beyond $O_2$.

## Methods

**Fabrication of various metal oxides on FTO**. The $BiVO_4$ precursor solution was prepared from a mixture of bismuth nitrate hexahydrate ($BiN_3O_9\cdot5H_2O$, 99.99%; Aldrich) and vanadyl acetylacetonate ($C_{10}H_{14}O_5V$, 98%; Aldrich), which were added to a solution of acetylacetone ($C_5H_8O_2$; Aldrich) and acetic acid ($CH_3COOH$, 99.70%; Fisher) with a ratio of 1:0.12, followed by sonication for 10 min. After sonication, a dark green solution was obtained and the solution is usually used within a day after the preparation. For the one named 1-layer, the mole concentration of Bi was varied from 0.08 M, and 0.04 M, 0.02 M, to 0.01 M. For a typical spin coating, 100 μl of the $BiVO_4$ precursor solution was dropped on a pre-cleaned FTO glass followed by spin coating (500 r.p.m. for 5 s and 1500 r.p.m. for 30 s). The samples were then annealed at 100 °C for 10 min, and 300, 400, and 500 °C for 5 min at each temperature. Similar step-wise annealing process was commonly used for metal oxide fabrication, and the purpose is to slowly evaporate the solvent to achieve a better film morphology. For spin coating multiple layers, the same process above was repeated for multiple times. Finally, the coated FTO was annealed in a box furnace at 500 °C for 2 h.

$SnO_2$ was fabricated by using a sol–gel process similar with $BiVO_4$. Firstly the precursor solution was made by dissolving 0.1932 g tin chloride in 10 ml 2-methoxyethanol ($CH_3OCH_2CH_2OH$, 99.8%; Aldrich) and 0.2 ml acetylacetone ($CH_3COCH_2COCH_3$, ≥99.3%; Aldrich) as the best condition, and sonicated for 30 min. After then the solution was put in a fume hood with aging for one night. The solution was spin coated on top of cleaned FTO with first 5 s, 500 r.p.m. and second 35 s, 2000 r.p.m. steps. Annealing process was carried out by using step by step method (100 °C for 5 min, 300 °C for 5 min and finally 455 °C for 30 min).

$TiO_2$ was fabricated from a paste making and coating process. About 9.5 ml ethanol and 0.5 ml water were mixed and 0.5 g polyethylene glycol (Aldrich) was added and the mixture was sonicated for 30 min. Then 0.25 g $TiO_2$ powder (Aldrich) as the best condition was added and the suspension was sonicated for another 10 min. Then the suspension was put on a hot plate and heated at 120 °C until the total volume reached 5 ml to get the $TiO_2$ paste. The paste was used to spin coat or blade coat on top of FTO glass, followed by the annealing process at 400 °C for 2 h.

$WO_3$ was synthesized by FVD method[46], for which a W wire (0.5 mm in diameter; Aldrich) was oxidized by flame as the WO$x$ vapor source. Those WO$x$ vapor further deposited on FTO as $W_{18}O_{49}$ nanowires. The optimized FVD condition followed the one used in Rao et al.'s work[46], with 18.4SLPM:12.5SLPM $CH_4$ to air flow ratio and a substrate temperature 550 °C for 10 min of deposition (Supplementary Table 4). $W_{18}O_{49}$ nanowires were further converted to $WO_3$ nanowires by annealing at 500 °C in a box furnace for 1 h.

**Characterizations**. The $H_2O_2$ production from various anodes was detected by using the potentiostat with the three-electrode system, in which the Ag/AgCl electrode was used as a reference electrode and Ti foil as the counter electrode. Silver paste and Teflon tape were used to make metal contact and to define area when necessary. The measurements under illumination were obtained under 1 Sun at AM 1.5 G. The amount of generated $H_2O_2$ was detected by using the standard $H_2O_2$ strips (Indigo Instruments). In addition, the generated $H_2O_2$ concentration was further confirmed with a titration process by using potassium permanganate ($KMnO_4$, ≥99.0%; Aldrich). The permanganate ion has a dark purple color, and the color disappears during titration when the $MnO_4^-$ is totally consumed based on the following equation:

$$2MnO_4^- + 5H_2O_2 + 6H^+ \rightarrow 2Mn^{2+} + 5O_2 + 8H_2O \tag{6}$$

In this work, the sulfuric acid ($H_2SO_4$; Acros Organics) was used as the $H^+$ source. We measured five $H_2O_2$ solution samples with different degrees of dilution from a same initial concentration 100 ppm, by both the standard strips and the permanganate titration methods. The results are shown in Supplementary Fig. 7, which shows that two methods basically agree with each other, confirming the accuracy of the $H_2O_2$ concentration measurement.

To make the result of the onset potential of $H_2O_2$ generation value more fair and accurate, in addition to the dark current onset here it was also defined as the potential at which the $H_2O_2$ generation starts to be detected (beyond 1 ppm), by measuring a 1 cm$^2$ sample in a 20 ml electrolyte for 10 min. $H_2$ and $O_2$ were detected by the gas chromatography analysis, and the morphology of the samples were obtained using scanning electron microscopy (FEI XL30 Sirion SEM). The FE for $H_2O_2$ production (%) is calculated by

$$FE = \frac{\text{Amount of generated } H_2O_2 \text{(mol)}}{\text{theoretical generated } H_2O_2} \times 100 \tag{7}$$

where the theoretical amount of $H_2O_2$ is equal to the total number of electrons divided by two (in mol). The FE of $H_2O_2$ is calculated based on the accumulated amount of $H_2O_2$ after the 10 min measurement for each condition. The FEs for $H_2$ and $O_2$ are calculated in a similar way, in which the theoretical amount of $H_2$ is also

equal to the total amount of electrons divided by two (in mol), while the theoretical amount of $O_2$ is equal to the total amount of electrons divided by four (in mol), respectively.

**Computational details**. Density functional theory calculations are done using the projector-augmented wave method and a plane-wave basis set as implemented in the Vienna Ab Initio Simulation Package (VASP). The valence configurations are treated as $6s^26p^3$ for Bi, $3d^34s^2$ for V, $2s^22p^4$ for O and $1s^1$ for H. The cutoff energy for plane-wave basis functions is 400 eV. The bulk and surface properties of $BiVO_4$ are optimized within GGA-PBE. For a more accurate description, the calculations are done within GGA-rPBE for the adsorption energies of OH*, O* and OOH* species on the $BiVO_4(111)$ surface. The reference energies of the pristine slab, $H_2$, $H_2O$ and $O_2$ molecules are also carefully treated within GGA-rPBE. For periodic slab calculations, slabs of six metal-oxygen layers are separated by at least 12 Å of vacuum. The atomic positions within the top two layers of the slabs were allowed to relax with the force convergence of 0.02 eV per Å. Spin polarization is considered in all the calculations.

**Data availability**. Data supporting the findings of this study are available within the article and its supplementary information files, and from the corresponding author upon reasonable request.

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

## Acknowledgements

We gratefully acknowledge support from the U.S. Department of Energy, Office of Sciences, Office of Basic Energy Sciences, to the SUNCAT Center for Interface Science and Catalysis. S.S. acknowledges support from the Global Climate Energy Project (GCEP) at Stanford University (Fund No.52454). G.L.L. acknowledges support from Henan University of Science and Technology (No. 2013ZCX018) and National Natural Science Foundation of China (Nos. U1404212 and 11404098). G.L.L. is grateful to the CSC scholarship.

## Author contributions

J.K.N. and S.S. conceived and designed the DFT calculations. X.Z. conceived and designed the experiment. S.S. and G.-L.L. performed the DFT calculation. X.S. performed the experimental measurement. S.S., X.S. and G.-L.L. prepared the manuscript. Y.Z. and P.C. assisted on the experiment. All authors discussed the results and commented on the manuscript.

## Additional information

**Competing interests:** The authors declare no competing financial interests.

