## [Peer Review File · Nature Communications]

Reviewers' comments:

Reviewer #1 (Remarks to the Author):

The authors investigated on the oxidative H₂O₂ production on four oxide/FTO anodes (WO₃, SnO₂, TiO₂ and BiVO₄) in the dark or under solar light. It is interesting to predict the electrochemical activity by DFT calculation. However, most of experimental findings are the same as those in the latest paper (K. Fuku, et al., Chemistry Select, 2016, 1, 5721) and Ref. 12. The fundamental idea of the comparison between theoretical limiting potential and the onset potential of oxide/FTO electrode is not right. I can not recommend this manuscript for the publication in NatureCommun.

1. Fuku, et al. recently reported the oxidative H₂O₂ production on 11 oxide/FTO anodes including WO₃, SnO₂ (=FTO), TiO₂, BiVO₄ in the dark in KHCO₃, and concluded that BiVO₄ is the best material among them (Chemistry Select, 2016, 1, 5721). They also reported BiVO₄/FTO photoanode is very efficient for the oxidative H₂O₂ production in KHCO₃ (Ref. 12). These results are completely same as the main experimental conclusion in this manuscript.

2. WO₃, SnO₂, TiO₂ and BiVO₄ are wide bandgap semiconductors and they act as insulator in the dark generally unless some defect level is present in the forbidden band. Therefore, the electron transfer may occur mainly on the FTO surface under the porous oxide layer or on the interface of oxide/FTO. The onset potential is defined by many factors such as coverage of insulator oxide, electro-catalytic activity at the interface of oxide/FTO, adsorption of cation/anion in the electrolyte solution. The onset potential is very changeable by the preparation methods of oxide/FTO. Both O₂ and H₂O₂ productions occur around the onset potential. The theoretical limiting potentials of oxides in this manuscript are calculated on the pure oxide surface. The electrolyte solution for WO₃/FTO was different from those for the others. The effect of carbonate anion is very important for the H₂O₂ production, but this effect is not considered in the DFT calculation. In conclusion, the fundamental idea of the comparison between theoretical limiting potential and the onset potential of oxide/FTO electrode is wrong.

3. High Faraday efficiency is interesting, but it is only the initial reaction in 10 min. The H₂O₂ will be re-oxidized or photo- or thermal-decomposed to O₂ easily. The accumulation data of H₂O₂ and Faraday efficiency after a long time are essential to utilize the H₂O₂ solution. High Faraday efficiency (92% - 99%) is already reported using Ge-oxyl complex photoanode, though the photocurrent properties and the accumulated amount of H₂O₂ was not high (T. Shiragami et al., Journal of Photochemistry and Photobiology A: Chemistry 313 (2015) 131).

4. The onset potential at 1.1 V vs RHE under solar light is not efficient compared to other previous BiVO₄ photoanode. Moreover, there is no photocurrent density-potential data. Why? It is probably very poor.

5. In Fig. 4, the Faraday efficiency was significantly changed by the potential. The author should explain the reason. Because the hole on the valence band in photoanode is hardly affected by the potential. The Faraday efficiency was also changed by the potential under the dark condition in Fig. 3, and the optimum potential for the maximum Faraday efficiency was different in each anodes. Why?

6. Page2-Line56: O₂ generation is Eq.4.

Reviewer #2 (Remarks to the Author):

This paper reports the DFT predictions predict trends in activity for water oxidation towards production of H₂O₂ on four different metal oxides i.e., WO₃, SnO₂, TiO₂ and BiVO₄. The trend for production of H₂O₂ was confirmed by experiments. BiVO₄ has provided the faraday efficiency about 95% for H₂O₂ production. The results are interesting enough to warrant publication in Nat. Commun. However, the following points should be clarified prior to publication.

- (1) The recent reports on photocatalytic production of H₂O₂ (Mase et al., Nat. Commun. 7, 11470 (2016); ACS Energy Lett. 1, 913-919 (2016)) should be cited.
- (2) The formation of H₂O₂ and the yield should be confirmed by using different methods. The yield of O₂ should also be determined.
- (3) The labeling experiments using H₂¹⁸O are also required to confirm that oxygen in H₂O₂ comes from O₂.

Reviewer #3 (Remarks to the Author):

The manuscript entitled "Understanding Activity Trends in Electrochemical Water Oxidation to Form Hydrogen Peroxide" demonstrates a theoretical study using DFT calculation to predict the activity trend of four representative metal oxides on photo-electrochemically hydrogen peroxide generation. Experimental investigation also confirmed the trend predicted by the DFT calculation. I believe the results are important for advancing our understanding of photo-electrochemical reactions. This topic is also expected to attract a broader readership as hydrogen peroxide generation is a important process in almost every water-related catalytic activity. The paper is suitable for publishing in Nature Communications after some technical issues in the manuscript should be addressed and clarified.

1. The introduction part is confusing. Information given in this part seems contradictory. For example, why does Eq.(2) represent a two-electron water oxidation process (page 2, line 49) but it is clearly a reducing process? Why is Eq. (5) an oxygen generation reaction while the product is not oxygen gas? In addition, no background was provided for the participation of OH*, O* and OOH* in this section, albeit they were frequently mentioned in the calculation section. The introduction part needs to be revised.
2. The rationale of choosing some specific crystal facets for the DFT calculation is unclear. If it is based on experimental observation, XRD and/or high-resolution TEM should be provided.
3. Information on how to evaluate the Faraday efficiency is missing.
4. The authors mentioned that the concentration of obtained hydrogen peroxide was experimentally determined by a titration process using potassium permanganate. Details (e.g., mechanism, accuracy, systematic errors, consistency comparing to the strip measurement etc.) should be provided. Please also comment on whether hydrogen peroxide can decompose during titration due to its chemical instability.
5. It is well-known that BiVO₄ is photo-electrochemically unstable. The authors should comment on the stability of BiVO₄ on hydrogen peroxide generation.
6. The rationale of the step-wise annealing process for material synthesis should be explained.

Point-by-Point response to the reviewers' comments (in blue)

Reviewer #1:

Comment 1

Fuku, et al. recently reported the oxidative H₂O₂ production on eleven oxide/FTO anodes including WO₃, SnO₂, TiO₂, BiVO₄ in the dark in KHCO₃, and concluded that BiVO₄ is the best material among them (Chemistry Select, 2016, 1, 5721). They also reported BiVO₄/FTO photoanode is very efficient for the oxidative H₂O₂ production in KHCO₃ (Ref. 12). These results are completely same as the main experimental conclusion in this manuscript.

Response:

We agree that Fuku et al. have published the pioneering work of using common metal oxides for oxidative H₂O₂ production, and our work falls into the same general area. However, we built upon their excellent work and went beyond their experimental scopes to achieve significantly higher performance for H₂O₂ production with complimentary theoretical efforts.

Specifically, Fuku's work (Chemistry Select, 2016) studied 11 oxides in a two-electrode system under a fixed bias of 3V. They studied BiVO₄ generation under a very large applied bias (3V-7V, vs. Pt mesh) with 1V interval, which is away from optimized condition for H₂O₂ production. We studied 4 oxides in a three-electrode system with a refined applied bias range and intervals for measurement points, to catch the optimal condition for all these four common used metal oxide. The variation of bias may appear trivial, but it is critical for investigating the competition among one-, two- and four-electron pathways under different bias. The elaborated study with different bias allows us not only to identify the optimal bias range for H₂O₂ production experimental, but also to compare with theoretical results on how these competing reactions vary with bias.

Because of our more systematic efforts on the bias dependence, as well as the different surface density controlling for dark and illumination, we have achieved much higher faraday efficiencies and H₂O₂ production rates under both conditions (see table above), in comparison to two papers by Fuku et al (which have been cited as the key references in the revised manuscript).

Comment 2

2. WO₃, SnO₂, TiO₂ and BiVO₄ are wide bandgap semiconductors and they act as insulator in the dark generally unless some defect level is present in the forbidden band. Therefore, the electron transfer may occur mainly on the FTO surface under the porous oxide layer or on the interface of oxide/FTO. The onset potential is defined by many factors such as coverage of insulator oxide, electro-catalytic activity at the interface of oxide/FTO, adsorption of cation/anion in the electrolyte solution. The onset potential is very changeable by the preparation methods of oxide/FTO. Both O₂ and H₂O₂ productions occur around the onset potential. The theoretical limiting potentials of oxides in this manuscript are calculated on the pure oxide surface. The electrolyte solution for WO₃/FTO was different from those for the others. The effect of carbonate anion is very important for the H₂O₂ production, but this effect is not considered in the DFT calculation. In conclusion, the fundamental idea of the comparison between theoretical limiting potential and the onset potential of oxide/FTO electrode is wrong.

Response:

We agree with the reviewer that our theory cannot duplicate the exact experimental conditions and experimental results are sensitive to the experimental details, a situation that is true whenever we compare any theories with experiments. This is exactly the reason that although our experiments appear to the same as Fuku's work in studying oxides for H₂O₂ production, our efforts on varying the applied bias are important. In addition, we strongly disagree with reviewer's view that comparing theory with experiment is not meaningful. Theory has played a vital role by providing insights on the nature of active sites and guiding the design and optimization of various catalysts (see the following references for example, Nørskov, J. K., Bligaard, T., Rossmeisl, J. & Christensen, C. H. Towards the computational design of solid catalysts. *Nat. Chem.* **1**, 37–46 (2009) and Seh, Z. W. et al. Combining theory and experiment in electrocatalysis: Insights into materials design. *Science* **355**, 4998 (2017)).

We disagree with the reviewer's comment regarding similar onset potential for both O₂ and H₂O₂ because thermodynamically the O₂ is evolved at 1.23 V while H₂O₂ is evolved at 1.76 V as can be seen in Figure 1 (in the revised manuscript). This difference in potential is reflected in the onset potentials and thus, production of O₂ and H₂O₂ occurs at different onset potential. Therefore, H₂O₂ evolution doesn't necessarily occur at the same bias when O₂ evolves. In fact, this is the main message of our study which we tried to highlight the principle for better H₂O₂ generation over O₂ generation. More information about this is covered in the comment Four.

We agree that the *J-V* curve onset is sensitive the interface of oxide/FTO, but this holds only if a large portion of FTO is exposed. In our experiments, we have optimized the coating of each oxide to minimize the exposure of FTO (Figure S4). Under those conditions, the onset potential is mainly affected by the surface properties of those oxides for O₂/H₂O₂ evolution. Our dark *J-V* curve onsets are very similar to those previous studies on BiVO₄ [1,2] and WO₃ [3,4] for water oxidation reactions, regardless of their precursor and/or fabrication method used. Moreover, we would like to clarify that we have studied the most stable facet of the oxides theoretically, however, we do not claim a one to one correspondence between the theoretical values and the experimental *J-V* onset for the studied oxides. Instead, we have focused on the trends (the hollow symbols in Figure 2b in the manuscript) in this work. Furthermore, we used not only *J-V* onset potential, but also the actual faraday efficiency of H₂O₂ production under different bias, to compare with our theoretical analysis (the solid symbols in Figure 2b in the manuscript), which is more meaningful since it reflects the real starting point for H₂O₂ evolution.

We want to clarify that the reason for choosing different electrolyte solution for WO₃/FTO is due to the poor stability of WO₃ in alkaline electrolyte (the as-prepared 1M NaHCO₃ solution has a pH 8.3). For WO₃, we still used the NaHCO₃ solution but adjusted its pH using an acid solution to solve the stability problem. We have considered the pH effect and normalized all the potentials to the ones vs. RHE.

Finally, we agree with the reviewer on the promotional effect of carbonate anion on H₂O₂ production, as Fuku's work suggested. This work focuses on the mechanism of H₂O₂ generation dependency on the surface properties for different materials as well as the applied bias, while the study of the anion effect is out of the scope of this work.

Comment 3

3. High Faraday efficiency is interesting, but it is only the initial reaction in 10 min. The H₂O₂ will be re-oxidized or photo- or thermal-decomposed to O₂ easily. The accumulation data of H₂O₂ and Faraday efficiency after a long time are essential to utilize the H₂O₂ solution. High Faraday efficiency (92% - 99%) is already reported using Ge-oxyl complex photoanode, though the photocurrent properties and the accumulated amount of H₂O₂ was not high (T. Shiragami et al., Journal of Photochemistry and Photobiology A: Chemistry 313 (2015) 131).

Response:

We agree that the accumulation data of H₂O₂ is important for practical H₂O₂ production, which is a system level optimization. This work aims to address the first step: identify suitable H₂O₂ production bias over metal oxides and understand the fundamental thermodynamic properties that affect the generation of H₂O₂. With those understandings, we achieved a high faraday efficiency for H₂O₂ production over BiVO₄. We have not studied the stability issue of H₂O₂ and are currently working on the accumulation and long-term storage of H₂O₂ obtained from the approach in this work.

Comment 4

4. The onset potential at 1.1 V vs RHE under solar light is not efficient compared to other previous BiVO₄ photoanode. Moreover, there is no photocurrent density-potential data. Why? It is probably very poor.

Response:

We didn't include the *J-V* curves not because they are poor but rather the current value includes both O₂ production and H₂O₂ production; therefore instead, we've used faraday efficiency of H₂O₂ that is more meaningful for H₂O₂ production. Per reviewer's request, we have added the *J-V* curve and the amount of H₂O₂ produced here. The onset potential of our BiVO₄ is well in the normal range as reported previously (0.47V-0.8V vs. RHE) [1, 2], and since this is in bicarbonate, the performance is even higher than the previous reports [1, 2, 5].

To address reviewer's point the following figure has been added as the new Figure S5 in the revised SI, and the following contents were added in the section "H₂O₂ production on BiVO₄ under different conditions" in the main text:

"The *J-V* curve and the measured H₂O₂ generation rate under illumination for this 9-layer BiVO₄ are shown as Figure S5".

Comment 5

5. In Fig. 4, the Faraday efficiency was significantly changed by the potential. The author should explain the reason. Because the hole on the valence band in photoanode is hardly affected by the potential. The Faraday efficiency was also changed by the potential under the dark condition in Fig. 3, and the optimum potential for the maximum Faraday efficiency was different in each anodes. Why?

Response:

The reviewer's comment is exactly the key point for this work: H₂O₂ production efficiency is bias dependent. This is a major difference between our work (varying bias) and Fuku's work (mainly focus on fixed bias). Having holes on the valence band is only a necessary, not sufficient, condition for H₂O₂ production. Optimizing H₂O₂ production needs to consider the competing O₂ generation at low bias and OH* formation at high bias, which leads to the bell shape faraday efficiency for H₂O₂ vs. applied bias. If only a rough and broad range is considered, the optimal condition may be missed since faraday efficiency under both higher or lower potential is low. Therefore, a refined potential range should be applied guided by the theoretical study. The competition between three reactions also depends on the materials. Our theoretical results (Figure 1) help us to understand the activity of different oxides in their relevant bias range, which is a very valuable contribution.

6. Page2-Line56: O₂ generation is Eq.4.

Response:

We thank the reviewer for the correction and corrected the text accordingly.

Reviewer #2

This paper reports the DFT predictions predict trends in activity for water oxidation towards production of H₂O₂ on four different metal oxides i.e., WO₃, SnO₂, TiO₂ and BiVO₄. The trend for

production of H₂O₂ was confirmed by experiments. BiVO₄ has provided the optimal faraday efficiency for H₂O₂ production. The results are interesting enough to warrant publication in Nat. Commun. However, the following points should be clarified prior to publication.

Comment 1

(1) The recent reports on photocatalytic production of H₂O₂ (Mase et al., Nat. Commun. 7, 11470 (2016); ACS Energy Lett. 1, 913-919 (2016)) should be cited.

Response:

We thank the reviewer for this comment and have cited the mentioned literatures. These two papers have given great contribution on the use of cobalt (II) chlorin complex as an effective catalyst for H₂O₂ production from O₂ reduction.

Comment 2

(2) The formation of H₂O₂ and the yield should be confirmed by using different methods. The yield of O₂ should also be determined.

Response:

We thank the reviewer for this comment and to address this point we have used KMnO₄ solution to further confirm the H₂O₂ yield. The details about the mechanism and accuracy on it are described in the “Methods” section as follows:

“The generated H₂O₂ concentration was further confirmed with a titration process by using potassium permanganate (KMnO₄, ≥99.0%, Aldrich). The permanganate ion has a dark purple color, and the color disappears during titration as the MnO₄⁻ being consumed based on the following equation:

In this work the sulfuric acid (H₂SO₄, Acros Organics) was used as the H⁺ source. We measured five H₂O₂ solution samples with different degree of dilution from a same initial concentration 100ppm, by both the standard strips and the permanganate titration are shown as Figure S7 (also shown below), which shows that two methods basically agree with each other, confirming the actuary of the H₂O₂ concentration measurement.”

The faraday efficiency for the yield of O₂ is also determined and added in the supplementary information as Figure S6 (shown in the Comment Three). In addition, the following contents are added in the sentence “H₂O₂ production on BiVO₄ under different conditions” in the main text: “In addition, we have measured the evolved gaseous O₂ and H₂ and the measured FEs are shown as Figure S6. Figure S6 shows the sum of the FE (O₂) and FE (H₂O₂) is about 98% to 103%, confirming the accuracy in our H₂O₂ concentration measurement.”

Comment 3

(3) The labeling experiments using H_2^{18}O are also required to confirm that oxygen in H_2O_2 comes from O_2 .

Response:

We thank the reviewer for this comment. To answer reviewer's point we would like to mention that in this work we've focused on the study of H_2O_2 generation from H_2O oxidation at the working electrode, other than from O_2 reduction at the counter electrode. We used a Nafion membrane to separate the working and counter and found that the yield of H_2O_2 at the counter is zero. In addition, the measured faraday efficiency for hydrogen is approximately 100% in a broad potential range, as shown in the Figure S6 (also shown below), which indicates all the electrons have been used to reduce H^+ to generate H_2 , instead of reducing O_2 .

Reviewer #3

The manuscript entitled “Understanding Activity Trends in Electrochemical Water Oxidation to Form Hydrogen Peroxide” demonstrates a theoretical study using DFT calculation to predict the activity trend of four representative metal oxides on photo-electrochemically hydrogen peroxide generation. Experimental investigation also confirmed the trend predicted by the DFT calculation. I believe the results are important for advancing our understanding of photo-electrochemical reactions. This topic is also expected to attract a broader readership as hydrogen peroxide generation is an important process in almost every water-related catalytic activity. The paper is suitable for publishing in Nature Communications after some technical issues in the manuscript should be addressed and clarified.

Comment 1

The introduction part is confusing. Information given in this part seems contradictory. For example, why does Eq.(2) represent a two-electron water oxidation process (page 2, line 49) but it is clearly a reducing process? Why is Eq. (5) an oxygen generation reaction while the product is not oxygen gas? In addition, no background was provided for the participation of OH*, O* and OOH* in this section, albeit they were frequently mentioned in the calculation section. The introduction part needs to be revised.

Response:

We thank the reviewer for this comment and pointing to typo error in the numbering of the equations in the text. To address this point, we have corrected the numbering of the equations in the text. We have also added the following sentence in the last line of the page 2 and the top of the page 3 to address reviewer's point regarding the background for OH*, O* and OOH*.

“The relevant intermediates of the one- (eq. 5), two- (eq. 1) and four-electron (eq. (4)) water oxidation reactions are OH*, O* and OOH*.”

Comment 2

The rationale of choosing some specific crystal facets for the DFT calculation is unclear. If it is based on experimental observation, XRD and/or high-resolution TEM should be provided.

Response:

We thank the reviewer for this comment. To answer reviewer's point we would like to mention that we have theoretically investigated the activity of all the low-index surfaces ((001), (101), (011), (111), (110), (100) and (010)) of fm-BiVO₄ (“Figure I” below). Among all the examined surfaces we found (111) and (110) as the most active surfaces for two-electron water oxidation (“Figure II” below). In fact (111) facet is among the most stable facets in terms of formation energies and it is highly likely to be an exposed facet in the crystal structure of the BiVO₄. The details of this analysis is out of the scope of this study and therefore we removed it from the SI in the first place. These results will be published separately. However, to address reviewer's point we have added the following sentence in the theoretical section in page 3.

“We only focus on the (111) surface, which has been shown theoretically and experimentally to be stable and exposed in the BiVO₄ crystal structure.³³”

Figure I. All the examined low-index surfaces of fm-BiVO₄. Side view: (a) (001), (b) (011), (c) (101), (d) (111), (f) (110), (g) (010) and (h) (100). Top view: (e) (111). Atoms underneath surface BiO_n ($5 \leq n \leq 7$) and VO₄ polyhedra are denoted by sticks. Adsorption of OH, O and OOH occurs on the Bi top sites at the (001), (011) and (101) surfaces. Adsorption of OH, O and OOH happens on the bridge sites between Bi1 and Bi4 as well as between Bi2 and Bi3 for the (111) surfaces, between Bi1 and Bi2 for the (110) surfaces, and between Bi1 and Bi1' for the (010) and (100) surfaces.

Figure II. Activity volcano plots based on calculated limiting potentials as function of calculated adsorption energies of OH^* (ΔG_{OH^*}) for the two-electron oxidation of water to hydrogen peroxide evolution (black) and the four-electron oxidation to oxygen evolution (blue). The corresponding equilibrium potentials for each reaction have been shown in dashed lines. Red circles indicate the activity of different surfaces of $BiVO_4$.

Comment 3

Information on how to evaluate the Faraday efficiency is missing.

Response:

We thank the reviewer for this comment. To address this point we have added the following explanation in the revised “Methods” section in the main text:

“The FE for H_2O_2 production (%) is calculated by

$$FE = \frac{\text{Amount of generated } H_2O_2 \text{ (mol)}}{\text{theoretical generated } H_2O_2} \times 100 \quad (7)$$

Where the theoretical generated H_2O_2 is equal to the total number of electrons divided by two (in mol);”

Comment 4

4. The authors mentioned that the concentration of obtained hydrogen peroxide was experimentally determined by a titration process using potassium permanganate. Details (e.g., mechanism, accuracy, consistency comparing to the strip measurement *etc.*) should be provided. Please also comment on whether hydrogen peroxide can decompose during titration due to its chemical instability.

Response:

We thank the reviewer for this comment. To address this point, we added the requested information as well as the related discussion in the revised “Methods” section:

“The generated H₂O₂ concentration was further confirmed with a titration process by using potassium permanganate (KMnO₄, ≥99.0%, Aldrich). The permanganate ion has a dark purple color, and the color disappears during titration as the MnO₄⁻ being consumed based on the following equation:

In this work the sulfuric acid (H₂SO₄, Acros Organics) was used as the H⁺ source. We measured five H₂O₂ solution samples with different degree of dilution from a same initial concentration 100ppm, by both the standard strips and the permanganate titration are shown as Figure S7 (also shown below), which shows that two methods basically agree with each other, confirming the accuracy of the H₂O₂ concentration measurement.”

In addition, from the consistency between two methods, as well as the linear relationship shown for the five points it can be seen within the 0-100ppm range (the range used in this work) the permanganate titration doesn't affect much on the H₂O₂ stability.

Comment 5

5. It is well-known that BiVO₄ is photo-electrochemically unstable. The authors should comment on the stability of BiVO₄ on hydrogen peroxide generation.

Response:

We agree that the photoelectrochemical stability of BiVO₄ is an issue due to the V⁵⁺ dissolution into solution [6,7], when pH is far from neutral conditions. However, as reported recently [8], BiVO₄ is quite stable in the near neutral region and our bicarbonate solution has a near neutral pH value (8.3 as measured). In fact, Fuku's work (*Chem. Commun.*, 2016, 52, 5406) shows that the WO₃/BiVO₄ has a very good stability for H₂O₂ production when also measured in bicarbonate solution. Based on this we've added some comments on the stability of BiVO₄ for H₂O₂ production in the text right before the "Conclusion" section:

“...the photoelectrochemical stability of BiVO_4 is known to be an issue when the electrolyte is far from neutral conditions because the V^{5+} tends to dissolve into solution.^{42,43} However, we used the bicarbonate electrolyte with a measured pH value of 8.3, so BiVO_4 is relatively stable in this near neutral region.⁴⁴”

Comment 6

The rationale of the step-wise annealing process for material synthesis should be explained.

Response:

The step-wise annealing process is commonly used when fabricating metal oxide films from the sol-gel process. A rapid temperature rise during annealing will cause the rapid boiling of the solvent, leading to poor film morphology. To address this point, we have added the following information in the “Methods” section.

“Similar step-wise annealing process was commonly used for metal oxide fabrication, and the purpose is to slowly evaporate the solvent to achieve a better film morphology.”

Reference

- [1] Efficient solar water splitting by enhanced charge separation in a bismuth vanadate-silicon tandem photoelectrode; Fatwa F. Abdi, Lihao Han, Arno H. M. Smets, Miro Zeman, Bernard Dam & Roel van de Krol; *Nature Communications* **4**, Article number: 2195 (2013)
- [2] Dynamics of photogenerated holes in undoped BiVO_4 photoanodes for solar water oxidation; Yimeng Ma, Stephanie R. Pendlebury, Anna Reynal, Florian Le Formal and James R. Durrant; *Chem. Sci.*, 2014, 5, 2964-2973
- [3] Improved Charge Separation in $\text{WO}_3/\text{CuWO}_4$ Composite Photoanodes for Photoelectrochemical Water Oxidation, Danping Wang, Prince Saurabh Bassi, Huan Qi, Xin Zhao, Gurudayal, Lydia Helena Wong, Rong Xu, Thirumany Sritharan, and Zhong Chen, *Materials* 2016, 9(5), 348
- [4] Vapor treatment of nanocrystalline WO_3 photoanodes for enhanced photoelectrochemical performance in the decomposition of water, Po-Tsung Hsiao, Liang-Che Chen, Tzung-Luen Li and Hsisheng Teng, *J. Mater. Chem.*, 2011, 21, 19402-19409.
- [5] Nanoporous BiVO_4 Photoanodes with Dual-Layer Oxygen Evolution Catalysts for Solar Water Splitting, Tae Woo Kim, Kyoung-Shin Choi, *Science*, 2014, 343, 990-994
- [6] Photoelectrochemical decomposition of water on nanocrystalline BiVO_4 film electrodes under visible light, Sayama, K, Nomura A, Zou Z, Abe R, Abe Y, Arakawa H (2003) *Chem Commun* 2908–9
- [7] From Molecules to Materials Pathways to Artificial Photosynthesis, Editors: Rozhkova, Elena A., Ariga, Katsuhiko, 2015
- [8] Solar water splitting BiVO_4 thin film photoanodes by sol-gel dip coating technique, S.Hilliard, D.Friedrich, S. Kressman, H. Strub, V. Artero, C. Laberty-Robert, DOI: 10.1002/cptc.201700003

REVIEWERS' COMMENTS:

Reviewer #1 (Remarks to the Author):

Editorial note: All further comments were to the Editor only; the Reviewer provided no further comments to the author.

Reviewer #2 (Remarks to the Author):

The paper has been well revised. I recommend publication of this paper.

Reviewer #3 (Remarks to the Author):

The authors have addressed my comments. I recommend publication.

Reviewer #4 (Remarks to the Author):

Understanding Activity Trends in Electrochemical Water Oxidation to form Hydrogen Peroxide

The paper describes electrochemical and photo-electrochemical oxidation of water to H₂O₂ mainly on BiVO₄. The paper has both a DFT based part and an experimental test of the performance in dark and under 1 sun. BiVO₄ show a high selectivity toward the two electron reaction both in the dark and under illumination.

This is a solid prove of that water can be oxidized to H₂O₂, it is however still far from real application. I see challenges in that regard related to the relatively high pH thus poor stability of H₂O₂ and the oxidation of H₂O₂ by surface OH*.

However, I believe that electrochemical H₂O₂ is a very interesting and timely subject and the present paper definitely provides novel insight into this field and I therefore recommend publication in Nature Comm.

The paper has already been under review and the authors have carefully answered all points raised by the previous reviewers.

I have a small comment to the authors.

I don't find that the criteria for a good catalyst are clearly shown in the paper. I guess that the idea is $2\text{OH}^* \rightarrow \text{H}_2\text{O}_2$, which should mean that the lowest possible GOH is $3.5\text{eV}/2$. There is probably not a strong binding side to the H₂O₂ volcano as the reaction cannot go via O*. The potential cannot be higher than the potential needed for $\text{H}_2\text{O} \rightarrow \text{OH}^* + \text{H}^+ + \text{e}^-$ as the further reaction to O* when becomes possible. It seems to me that an important criterion is that the oxide is a bit off the relation between OH* and O*, which also seems to be the case looking at the OH* and O* energies reported in sup mat.

Response to Reviewers' comments:

Reviewer #4 (Remarks to the Author):

Understanding Activity Trends in Electrochemical Water Oxidation to form Hydrogen Peroxide

The paper describes electrochemical and photo-electrochemical oxidation of water to H₂O₂ mainly on BiVO₄. The paper has both a DFT based part and an experimental test of the performance in dark and under 1 sun. BiVO₄ show a high selectivity toward the two electron reaction both in the dark and under illumination.

This is a solid prove of that water can be oxidized to H₂O₂, it is however still far from real application. I see challenges in that regard related to the relatively high pH thus poor stability of H₂O₂ and the oxidation of H₂O₂ by surface OH*.

However, I believe that electrochemical H₂O₂ is a very interesting and timely subject and the present paper definitely provides novel insight into this field and I therefore recommend publication in Nature Comm.

The paper has already been under review and the authors have carefully answered all points raised by the previous reviewers.

Comment 1

I have a small comment to the authors.

I don't find that the criteria for a good catalyst are clearly shown in the paper. I guess that the idea is $2\text{OH}^* \diamond \text{H}_2\text{O}_2$, which should mean that the lowest possible GOH is $3.5\text{eV}/2$. There is probably not a strong binding side to the H₂O₂ volcano as the reaction cannot go via O*. The potential cannot be higher than the potential needed for $\text{H}_2\text{O} \diamond \text{OH}^* + \text{H}^+ + \text{e}^-$ as the further reaction to O* when becomes possible. It seems to me that an important criterion is that the oxide is a bit off the relation between OH* and O*, which also seems to be the case looking at the OH* and O* energies reported in sup mat.

We truly appreciate the referee for this comment and fully agree with this point that anything in the strong OH* binding regime cannot make H₂O₂ unless it deviates from the scaling relation. We have discussed this point in the following publication recently (Siahrostami, S., Li, G.-L., Viswanathan, V. & Nørskov, J. K. One- or Two-Electron Water Oxidation, Hydroxyl Radical, or H₂O₂ Evolution. *J. Phys. Chem. Lett.* 1157–1160 (2017)) where, we put all the required criteria for O* and OH* binding energies in a selectivity map for H₂O₂, O₂ and OH radical evolution (following Figure taken from the mentioned reference). It is indeed true that based on thermodynamics criteria, a large window for H₂O₂ evolution (shaded in green, ΔG_{O^*} above ~ 3.5 eV and ΔG_{OH^*} below ~ 2.4 eV) is expected. However, in practice, due to the scaling relation between the binding energies of O* and OH*, selective H₂O₂ evolution is limited to the lower right corner of the green window. As can be seen all

the oxides we have studied in the present work fall in the scaling line and they show selectivity towards H_2O_2 evolution, because their binding energies are in the small corner of the H_2O_2 selectivity space. This analysis supports the referee's point and shows there is room to find selective H_2O_2 catalysts that largely deviate from O^* and OH^* scaling.

To address referee's point, we have added the following two sentences in the manuscript, page 5, first paragraph (highlighted in yellow).

“Hence, the combined thermodynamic criteria and scaling relation indicates a selective catalyst for H_2O_2 evolution should have ΔG_{OH^*} from ~ 1.6 to 2.4 eV.”

“To increase the selectivity region for H_2O_2 evolution, we need to identify catalyst materials that largely deviate from the O^* and OH^* scaling relation.⁴²”

Figure is adapted from Ref. (*J. Phys. Chem. Lett.* 1157–1160 (2017)): Phase diagram in terms of the binding energies of O^* vs. OH^* . The black solid line displays the scaling line between O^* and OH^* on different oxides. Blue, green and red highlighted colors indicate regions in which O_2 , H_2O_2 or OH radical are expected to be the major product, respectively in terms of purely thermodynamic constraints.